# Inflammation and Organic Cation Transporters Novel (OCTNs)

**DOI:** 10.3390/biom14040392

**Published:** 2024-03-25

**Authors:** Lorena Pochini, Michele Galluccio, Lara Console, Mariafrancesca Scalise, Ivano Eberini, Cesare Indiveri

**Affiliations:** 1Laboratory of Biochemistry, Molecular Biotechnology and Molecular Biology, Department DiBEST (Biologia, Ecologia, Scienze della Terra), University of Calabria, Via Bucci 4C, 6C, 87036 Arcavacata di Rende, Italy; michele.galluccio@unical.it (M.G.); lara.console@unical.it (L.C.); mariafrancesca.scalise@unical.it (M.S.); 2Institute of Biomembranes, Bioenergetics and Molecular Biotechnologies (IBIOM), National Research Council (CNR), Via Amendola 122/O, 70126 Bari, Italy; 3Department of Pharmacological and Biomolecular Sciences, Università degli Studi di Milano, 20133 Milan, Italy; ivano.eberini@unimi.it

**Keywords:** membrane transporters, SLC, antioxidant, cation transporters, carnitine, acetylcholine

## Abstract

Inflammation is a physiological condition characterized by a complex interplay between different cells handled by metabolites and specific inflammatory-related molecules. In some pathological situations, inflammation persists underlying and worsening the pathological state. Over the years, two membrane transporters namely OCTN1 (SLC22A4) and OCTN2 (SLC22A5) have been shown to play specific roles in inflammation. These transporters form the OCTN subfamily within the larger SLC22 family. The link between these proteins and inflammation has been proposed based on their link to some chronic inflammatory diseases such as asthma, Crohn’s disease (CD), and rheumatoid arthritis (RA). Moreover, the two transporters show the ability to mediate the transport of several compounds including carnitine, carnitine derivatives, acetylcholine, ergothioneine, and gut microbiota by-products, which have been specifically associated with inflammation for their anti- or proinflammatory action. Therefore, the absorption and distribution of these molecules rely on the presence of OCTN1 and OCTN2, whose expression is modulated by inflammatory cytokines and transcription factors typically activated by inflammation. In the present review, we wish to provide a state of the art on OCTN1 and OCTN2 transport function and regulation in relationships with inflammation and inflammatory diseases focusing on the metabolic signature collected in different body districts and gene polymorphisms related to inflammatory diseases.

## 1. Introduction

Inflammation is a multistep and complex physiological reaction of the host to infections and damage. The purpose of inflammatory responses is to restore body homeostasis even though prolonged and unresolved inflammatory states are responsible for a plethora of pathological conditions with a broad range of severity. In the last few years, an interesting link between inflammation and metabolism has been suggested due to the increased need for nutrients from inflammatory cells [1]. In this scenario, a close relationship between inflammatory processes and membrane transporters has been described, in line with the role of these proteins in allowing for the flux of nutrients across tissues. Another relevant issue is the traffic of compounds that are produced or whose levels change during the inflammatory processes. Therefore, membrane transporters which can interact with or transport these inflammation-related compounds represent interesting targets for therapy in a wide range of diseases directly or indirectly related to inflammation. Among transporters exhibiting the property of interacting with many different compounds, there are the human Organic Cation Transporters Novel OCTN1 (SLC22A4) and OCTN2 (SLC22A5). These two proteins correspond to the A4 and A5 members of the Solute Carrier (SLC) family 22, which includes a large number of cations, zwitterions, and anion transporters. The cation and zwitterion transporter members share significant structural features [2,3]. OCTN1 and OCTN2 constitute a small subgroup sharing more than 76% identity to each other, which would indicate similar, or at least related, cellular roles and/or biochemical functions [4]. Notwithstanding, differences in the function of the two human transporters emerged from many studies performed in the last two decades [5,6]. Ergothioneine and acetylcholine have been identified as the major OCTN1 substrates, whereas carnitine and carnitine derivatives have been identified as the major substrates for OCTN2 [6,7,8,9,10,11,12,13,14]. Interestingly, ergothioneine was also reported to be transported by OCTN2 [15], and carnitine was reported as a low-affinity OCTN1 substrate more than 20 years ago [16]. Very recently, this finding has been confirmed by in vitro experiments showing that OCTN1 catalyzes a sodium-dependent carnitine transport [17]. Moreover, similarly to the OCT members of the SLC22 family [18,19], OCTN1 and 2 share polispecificity towards ligands including molecules related to the inflammatory processes (Figure 1).

Based on metabolomics, it has been found that small molecules that are enriched in an inflamed status in plasma, urine, synovial fluid, feces, or different tissues are typical ligands of SLC22 members and, especially, of OCTNs [18,19,22,23].

In particular, the transport of carnitine, acetylcarnitine, and ergothioneine mediated by OCTN1 occurs mainly by an uptake mode, whereas the transport of acetylcholine occurs by an efflux mode. In the case of the OCTN2 transport of carnitine or its analogs, it occurs by an uptake mode.

Circulating metabolites are conditioned by many factors, such as the diet–microbiome axis as well as by the activity of many enzymes and transporters. Indeed, metabolic and immune signals of the microbiota can enter circulation exploiting the activity of intestinal membrane transporters. In this context, OCTN1 and OCTN2 may also affect the level of these metabolites which are known (or still unknown) substrates of these transporters; moreover, the inflammatory processes may act directly or indirectly on the expression of OCTN1 and OCTN2 [24], thus influencing the level of the metabolites. In good agreement, altered levels of metabolites, which are known or possibly unknown OCTN substrates, have been found in the case of chronic inflammatory diseases, such as Inflammatory Bowel Diseases (IBDs) [23]. The similarity of the AlphaFold structures of the two proteins, shown in Figure 2A, correlates well with the sequence similarity between OCTN1 and OCTN2 (Figure 2B) and also with the Cryo-EM 3D structures of OCT1, 2, and 3 sharing from 31% to 35% identity with the OCTNs [25,26]. The predicted OCTN1-2 3D structures highlight the presence of a large extracellular loop with potential N-glycosylation sites. Moreover, the OCTNs share the presence of an intracellular nucleotide-binding domain which is different from the NBDs of the ABC transporters [27,28,29,30]. This is in line with the described regulation of OCTN1 by intracellular [31] or intraliposomal ATP [9,10,11,32].

The external loops of OCTN1 and OCTN2 contain four cysteines arranged in two couples of vicinal residues that can form disulfides (Figure 2).

This feature makes OCTNs sensitive to thiol reagents. As an example, toxic mercury derivatives can modify the function of these transporters at concentrations close to those found in contaminated environments [33,34]. In the case of OCTN1, the Cys residues responsible for the response to thiol-reactive compounds, such as mercury compounds, cysteine, N-acetylcysteine, and carboxymethyl cysteine, have been identified by site-directed mutagenesis [33,34]. It is then likely that OCTNs may interact with physiological thiol-reacting compounds such as Reactive Oxygen Species (ROS), thus responding to redox signaling [35]. These features correlate well with the above-mentioned involvement of the OCTNs in inflammatory processes [36,37,38,39], characterized, among other factors, by excess ROS formation [40]. OCTN1 and OCTN2 also share localization in tissues such as the lungs, gut epithelia, and immune cells, directly or indirectly involved in inflammatory processes [5,6,21,41,42,43,44,45]. Indeed, the frontier epithelia are exposed to microorganisms, inhaled drugs, cigarette smoke, and other pollutants that may give rise to inflammation [21,44,46]. OCTN1 also regulates the activation of microglia which are macrophages resident in the central nervous system and responsible for initiating innate immune responses to a variety of different pathogens. Microglial cells activate and migrate to the damaged regions, where the production of inflammatory cytokines occurs (IL-1β and TNFα, ROS, and neurotrophic factors). The chronic over-activation of microglia damages neurons as well, resulting in the onset of neurodegenerative disorders, such as Alzheimer’s and Parkinson’s diseases [47].

All the described OCTN features strongly support a role in inflammatory diseases and host defense mechanisms. Indeed, several reports indicate the involvement of OCTN1 and 2 in asthma, CD, and RA [5,6,45,48]. However, the molecular bases of the described link are still poorly defined. OCTNs could exert a role in the intestinal absorption of inflammation-linked metabolites, their distribution to tissues, and kidney excretion, also playing a role in cell communication processes. Interestingly, OCTN2 has been found as novel exosome cargo whose level in these endosomal-derived nano-vesicles is regulated by the proinflammatory cytokine INF-γ [24]. In this review, we will provide an overview of the relationships between inflammatory processes and OCTNs.

## 2. OCTN Functions and Dysfunctions

### 2.1. Relationships between Functions and Diseases

OCTN2 has a well-recognized role as a sodium-dependent carnitine transporter in most tissues. This function is fundamental for the accomplishment of the mitochondrial β-oxidation pathway that has an essential requirement for carnitine. In this frame, OCTN2 plays two important roles: (i) mediating carnitine absorption from diet, accounting for more than half of the carnitine body pool to compensate for the limited biosynthesis; (ii) distributing carnitine to most tissues that cannot synthesize it. Besides the well-established role of carnitine in the mitochondrial carnitine shuttle, other functions have also been demonstrated or proposed [49,50,51,52,53]. In the case of OCTN1, the major substrates are ergothioneine and acetylcholine, which are also related to antioxidant and/or anti-inflammatory effects. Ergothioneine is a mushroom metabolite known for its antioxidant activity [54]. Acetylcholine, differently from ergothioneine, is a physiological compound. Interestingly, acetylcholine, besides the well-known neurotransmitter function, also plays an anti-inflammatory role [55,56,57]. Acetylcholine is a ubiquitous signaling molecule produced by numerous non-neuronal cells that possess the same acetylcholine biosynthesis pathway as neurons [9,58,59]. The only difference is that non-neuronal cells do not excrete the neurotransmitter by a quantal (vesicular) mechanism but by a slow efflux mediated by transporters of the SLC22 family, among which is OCTN1 [10,60]. It has been known for 20 years that OCTN1 can transport carnitine as well, although with an affinity about two orders of magnitude lower than that of OCTN2. Very recently, this OCTN1 feature has been better clarified, confirming that OCTN1 mediates carnitine transport with a similar mechanism to OCTN2 [17]. However, the molecular basis of the difference in affinity is still a matter of investigation. Interestingly, some physio-pathological observations correlate well with the described similarities and differences of the two proteins with respect to carnitine. The defects of OCTN2 are causative of Primary Carnitine Deficiency (PCD), a severe syndrome characterized by progressive skeletal muscle weakness and cardiomyopathy. During infancy, it may cause hypoketotic hypoglycemia and Reye syndrome, characterized by encephalopathy, hyperammonia, and sudden infant death [6,61,62]. However, PCD is not lethal, especially if treated with carnitine administration, differently from Secondary Carnitine Deficiency, caused by defects of the mitochondrial carnitine transporter [63]. The reason for the lack of lethality is the presence of alternative low-affinity carnitine absorption pathways. Therefore, owing to the overlapping localization of OCTN1 and OCTN2 and the ability of OCTN1 to transport carnitine, it is not trivial to assume that OCTN1 represents the main alternative pathway. Moreover, the involvement of OCTN2, and partially of OCTN1, in the fatty acid β-oxidation, constitutes a link between the two transporters with ROS. Indeed, it is well known that fatty acid catabolism contributes to ROS formation, and in the case of an excess of these reactive species, such as ischemia–reperfusion, damaged tissues benefit from the inhibition of fatty acid catabolism, as an example, by mildronate, a strong inhibitor of OCTN2 [64]. In line with the apparent accessory role of OCTN1 in fatty acid catabolism, the knockout of OCTN1, at least in mice, does not show any phenotype even though it exhibited greater susceptibility to intestinal inflammation under the ischemia and reperfusion model [65]. Differently from OCTN2, mutations of OCTN1 associated with a loss of function in humans are not known, and we cannot ensure that a hypothetical knocking out in humans would also not show any phenotype. Indeed, the mouse model is not considered a suitable model for humans anymore [6], considering the presence of a further transporter of the subfamily (OCTN3) that is absent in humans [5,16].

### 2.2. OCTN Polymorphisms and Relationships with Pathologies

So far, some OCTN polymorphisms have been described which are associated with pathologies characterized by inflammation status (Appendix A). rs1050152 is the most well described OCTN1 polymorphism, leading to L503F amino acid substitution. This is associated with CD, an IBD that causes chronic inflammation of the gastrointestinal tract. Both OCTN genes, SLC22A4 and SLC22A5, are located within the IBD5 locus on chromosome 5. This region is implicated in susceptibility to IBDs, and OCTN variants have been associated with IBDs [48,66] (Appendix A). An SNP located in a Runt-related transcription factor 1 (RUNX1)-binding sequence in SLC22A4 has been found to affect the expression of OCTN1 by altering RUNX1 binding affinity [37]. The number of variants described for the SLC22A5 gene in the Variant Viewer tool of the UniProt database (which includes data deriving from different genome databases and bioinformatics resources among which are gnomAD, dbSNP, and TOPMed) is remarkably higher, counting 685 missense variations, involving 413 out of 557 amino acids (74%). Among these protein variants, 145 are pathogenic, most of which, 103, are associated with renal carnitine transport defects, and the other 38 are referred to as likely pathogenic. Among the pathogenic variants, 73 protein variants cause PCD with the loss of carnitine transport. The number of variants described in the Variant Viewer tool for the SLC22A4 gene counts 422 missense variations. These variations result in 298 out of the 557 amino acids of the OCTN1 protein sequence (54%) which have been found to be mutated. Appendix A reports all the references to the SNPs.

## 3. OCTNs’ Role in Inflammation

### 3.1. Involvement of OCTN Substrates in Inflammatory Processes

As already mentioned, a role in the inflammatory process has been reported for the major substrates of OCTNs [8,9,10,11,12,13,67,68].

Indeed, acetylcholine is known to regulate the expression of inflammatory cytokines in microglial cells [69]. Moreover, acetylcholine is involved in controlling inflammation via the non-neuronal cholinergic system by acting on the alpha 7 nicotinic receptors. Interestingly, OCTN1 can mediate a low-rate acetylcholine efflux from cells [11], which may be the basis of this physiological role.

Another OCTN substrate, ergothioneine, may be involved in the suppression of the inflammatory cytokine IL-1β expression. This effect is based on the antioxidant property of ergothioneine [47,70].

The common OCTN substrate, carnitine, has been acknowledged for its involvement in inflammation [71]. This became clear by observing the juvenile visceral steatosis (OCTN2−/−) mouse, which is a model of systemic carnitine deficiency. These mice developed intestinal villous atrophy, inflammation, ulcer formation, and gut perforation [72]. In these mice, the downregulation of the TGF-beta/BMP pathway has been observed [73]. Subsequently, other evidence confirmed the anti-inflammatory, immunosuppressive, and therapeutic properties of carnitine [74]. Carnitine proved to be effective in suppressing lipopolysaccharides (LPS)-induced cytokine production and improving murine survival rates during cachexia and septic shock. This substrate also exerts an inhibitory effect on inducible nitric oxide synthase (iNOS) and, hence, on nitric oxide (NO) production, reducing inflammation and histological damage in murine trinitrobenzene sulphonic acid-induced colitis [52,75]. Carnitine involvement in inflammation could also be correlated to the metabolic reprogramming from glucose to fatty acid oxidation, occurring during inflammatory responses [76,77]. Indeed, carnitine would play a critical role in the activation of M2 macrophages and inflammasome activation in M1 macrophages. However, the role of fatty acid oxidation in macrophage polarization (towards M1 or M2) and the molecular mechanism behind this process are still controversial [77]. Interestingly, the activation of the oxidative program in M2 is mediated by the activation of the Peroxisome proliferator-activated receptor (PPAR) γ, the transcription factor involved in regulating lipid metabolism, energy homeostasis, and OCTN transcription (see Section 4).

Anti-inflammatory effects of the carnitine derivatives are also reported. Acetyl-L-carnitine exerts a role in LPS-induced neuroinflammation in rats, by targeting the TLR4/ nuclear factor kappa-light-chain-enhancer of the activated B cells (NFκB) pathway [78]. In a valproate model of autism, chronic treatment with acetyl-carnitine alleviated behavioral abnormalities through different mechanisms including the recovery of inflammation in the brain [79]. OCTN2 expression in the blood–brain barrier (BBB) [80] would facilitate the permeation of carnitine and acetyl-carnitine to the brain tissue [81]. This makes carnitine and carnitine derivatives therapeutic candidates for all those neurological diseases in which inflammation represents a hallmark, such as Parkinson’s and Alzheimer’s diseases. The effects of carnitine supplementation on inflammatory markers have been reported and investigated in randomized controlled trials as well [82,83]. Experiments performed in atherosclerotic rats revealed a decrease in the level of mRNA and protein of CPR, TNF-a, IL-1b, and iNOS in the aorta and heart tissues as a consequence of acetyl-carnitine treatment [84]. The supplementation of propionyl-carnitine has been found to improve clinical response in ulcerative colitis (UC) [85,86].

The metabolism of choline, another substrate/ligand of OCTN1 [11], has been linked to the control of NLRP3 inflammasome-dependent inflammation [87]. Moreover, it has been observed that in the serum of two rodent models of Alzheimer’s disease, the levels of circulating choline were reduced, while proinflammatory cytokine TNFα was elevated [88].

In methionine-choline deficiency (MCD)-induced Non-Alcoholic Fatty Liver Disease (NAFLD), the OCTN2 substrate betaine [89] might reduce liver inflammation and damage, at least partly, by improving the balance between proinflammatory (TNF, IL-6) and anti-inflammatory (IL-10, TGF-β) cytokines [90].

Taken together, the reported data highlight an overall anti-inflammatory effect mediated by OCTN substrates.

### 3.2. OCTN Substrates and Gut Microbiota Communication/Interconnections

In a frontier district, i.e., facing the external environment, such as the gut, many of the OCTN2 and OCTN1 substrates may encounter a different fate: indeed, in gut lumen, microbiota may compete with the intestine epithelium for the absorption of the OCTN substrates [91,92]. The ammonium groups of choline, carnitine, ergothioneine, and betaine are converted into trimethylamine (TMA) by the gut microbiota [93]. TMA is absorbed through the intestine and oxidized by the liver enzymes monooxygenases in trimethylamine oxide (TMAO) [94]. Growing experimental evidence in animal models demonstrates the contribution of TMAO to inflammation [95], via the transcription factor NF-κB, which, in turn, triggers cytokine production [96]. Therefore, in this pathway, intestinal carnitine would have a role in TMAO production and inflammation. However, it has to be stressed that the absorption of carnitine by the intestine is a fast and efficient process due to the concentrative mechanism of transport resulting from the cooperation of OCTN2 and OCTN1 above-described in terms of carnitine transport [17]. Altogether, these observations indicate that most of the carnitine derived from a normal diet could be absorbed by the intestine. We then hypothesize that the possible indirect proinflammatory role of carnitine and/or choline could occur only under conditions of over-administration or defects of intestinal absorption, i.e., some pathological states [97]. This could in part explain the TMAO-mediated cardiomyopathy exacerbation in PCD, which is caused by mutations of OCTN2 (see Appendix A). Furthermore, carnitine would mediate the cross-talk between the microbiota and intestinal epithelium; a microbial metabolite, butyrate, is the primary metabolic fuel of the colonic epithelial cells: its oxidation, to which carnitine contributes, would provide colonocytes with 70% energy [75].

Other OCTN substrates contribute to the cross-talk between microbiota and intestinal cells. The intestinal microbiota adopts cholinergic metabolism as well [98]. In agreement, the release of epithelial acetylcholine, which is a substrate of OCTN1, is stimulated when the short-chain fatty acid (SCFA) receptor GPR41 (FFA3) and/or GPR43 (FFA2) bind propionate, another microbial metabolite produced during the fermentation of carbohydrates in the lumen of the large intestine. Microbes create total luminal concentrations of SCFA in the range of 100 mM by the fermentation of carbohydrates. Besides acetylcholine, the colonic mucosa can produce atypical choline esters such as propionylcholine and butyrylcholine due to the relative unspecificity of the choline acetyltransferase (ChAT) towards the SCFA which is esterified with choline; these choline derivatives may also be substrates of OCTN1, as it is known for choline. These atypical esters act on cholinergic epithelial receptors, too, but with a much lower affinity, so they are thought to modify epithelial cholinergic signaling by the desensitization of cholinergic receptors against acetylcholine. Differences in the SCFA profile should, therefore, affect the efficiency of the non-neuronal system with implications for the communication between intestinal microbiota and the mammalian host, modulating immunity and inflammation. In line with these observations, bacteria fermenting fibers and producing SCFAs are reduced in the mucosa and feces of patients with IBD [99]. Based on the described experimental evidence, it is clear that OCTN1 and OCTN2 are important players in the biochemical communication between microbiota and epithelial cells through the traffic of their substrates.

## 4. Regulation of OCTN Expression and Inflammation

### 4.1. Major Players of OCTN Regulation

SLC22A4 and SLC22A5 are target genes of the PPAR family. The members of the PPAR family (PPARα, PPARγ, and PPARβ/δ) are transcription factors involved in the regulation of lipid metabolism and energy homeostasis [100,101]. Among the plethora of functions controlled by these receptors, evidence suggests that all three PPAR subtypes play a significant role in controlling inflammatory responses [102] and in IBD [103]. A ligand-activated PPAR assembles into a complex with retinoid X receptor α (RXRα) and, as such, translocates to the nucleus. The anti-inflammatory efficacy of PPAR ligands is based on the PPAR/RXR-mediated blockade of the nuclear translocation of NF-kB, resulting in the transcriptional blockage of inflammatory cytokines, chemokines, and other stress response elements, such as cyclooxygenase-2 (COX2) and iNOS [103]. PPARs can control gene expression by associating with activator proteins or by binding Peroxisome Proliferator Responsive Elements (PPREs) on DNA. In the case of OCTN2, PPREs have been found within the promoter [104,105]. Moreover, the estrogen receptor is another regulator of OCTN2 expression [14] due to the presence of an estrogen receptor-responsive element in the SLC22A5 first intron [106]. Further transcriptional regulation involves the binding of a heat–shock element to the SLC22A5 promoter [36]. Unlike the above-described mechanisms, the methylation of the SLC22A5 promoter is responsible for its transcriptional downregulation [107]. The same epigenetic regulation has been observed as responsible for reduced OCTN1 expression [108]. The OCTN1 promoter region includes several other consensus recognition sites for ubiquitously expressed transcription factors, such as Sp1, RUNX1, and NF-kB [109]. Sp1 would be involved in the regulation of the tissue-specific expression of OCTN1 and not in its basal transcriptional regulation. RUNX1 has been found to be associated with OCTN1 transcriptional regulation: it functions both to activate and to repress transcription through interactions with cofactors [37] and is not associated with the basal promoter activity of OCTN1 [109]. Human fibroblasts derived from RA patients have been employed to investigate the regulation of SLC22A4 and SLC22A5 expression in the context of inflammation. In these cells, OCTN1 has been identified as a susceptibility gene for RA being regulated by RUNX1 [37]. NF-kB activation was observed when an agonist of the Toll-like receptor 3 (TLR3), polyinosinic–polycytidylic acid (poly I:C), was used to stimulate viral inflammation in rats. In this context, the downregulation of mRNA levels in Octn1 and Octn2 was found [110]. Concerning OCTN2, its mRNA was affected by neither IL-1beta nor TNFα; levels of OCTN1 mRNA were increased by stimulation with TNFα. IL-1beta nor TNFα are associated with OCTN1 transcriptional regulation via the NF-kB signaling cascade: the over-expression of NF-kB would activate the promoter activity of OCTN1. The increased OCTN1 level in inflammatory conditions would explain the increased uptake of the tyrosine kinase inhibitor, saracatinib, considering that hOCTN1 was identified as the main transport system for the accumulation of this TKI in RASF [111]. Concerning OCTN2, the involvement of NF-κB in the TNFα stimulation of the OCTN2 gene expression has been demonstrated in the Madin-Darby bovine kidney (MDBK) cell line [112].

### 4.2. OCTN Regulation in Immune System

Leukocytes (monocytes), intestinal immune cells, alveolar macrophages, and microglia are some of the immune cell types where higher OCTN expression levels have been found [20,37,47,68]. As a result of stimuli from pathogens, monocytes are recruited from the circulation and differentiate into macrophages (macrophage differentiation). OCTNs emerge as novel markers of this process (Figure 3).

Indeed, undifferentiated monocytes express high protein levels of OCTN1 that dramatically drop along with the differentiation into macrophages; the OCTN2 protein, undetectable in monocytes, became strongly expressed in monocyte-derived macrophages, with a 20-fold increase in the corresponding mRNA [74]. This process involves changes in gene expression driven by multiple transcription factors, among which are those related to the PPAR and STAT (signal transducers and activators of transcription) families. Monocyte/macrophage development is mainly influenced by the monocyte colony-stimulating factor (also known as CSF-1) and by the cytokine Granulocyte-Macrophage Colony-Stimulating Factor (GM-CSF). In particular, OCTN2 is induced during the differentiation of monocytes to macrophages by the mTOR-STAT3 pathway after a GM-CSF stimulus and not by PPAR transcription factors: GM-CSF causes the activation of mTOR kinase, leading to the phosphorylation and activation of the transcription factor STAT3, which, in turn, is responsible for OCTN2 transcription. Rapamycin, the specific inhibitor of mTOR, reduced the expression of the transporter at both mRNA and protein levels. Other activators of STAT3, like IL-6 and CSF-1, leading to the phosphorylation of STAT3, induced OCTN2 expression. Concerning CSF-1, a comparable induction of OCTN2 mRNA and protein is also observed upon the differentiation of monocytes. OCTN1, which is highly expressed in CD14+ monocytes, is downregulated by CSF1 [113]. Recently, a study proposed that the dysregulation of monocyte adaptation to the environment of the gastrointestinal mucosa is the key process leading to IBD. In the context of colitis, an increase in OCTN1 gene expression after the differentiation of THP-1 monocytic leukemia cells into macrophages using phorbol-12-myristate-13-acetate has been observed [68].

### 4.3. OCTN Regulation in Epithelia

Many of the findings concerning OCTN regulation in inflammation have been obtained focusing on the respiratory and gastrointestinal epithelia, which, although distinct, are considered part of a shared mucosal immune system, the “gut–lung axis”.

#### 4.3.1. Airway Epithelium

In the context of the airway epithelium, contradictory data were collected, and some of the most recent examples are reported here. An impressive increase in OCTN2 mRNA levels and OCTN1 and 2 protein expression was observed if an inflammatory reaction (representative of epithelial inflammation in asthma) was induced by exposing to microbe-specific stimulus LPS or HDM the ALI Calu-3 layers, an in vitro model anatomically similar to the native bronchial epithelium [114]. In contrast, a lack of the regulation of the OCTN2 expression levels by LPS has been observed in a further study performed in Calu-3 cells and in the EpiAirway™ system, which consists of a pseudostratified epithelium containing the differentiated cell types found in the respiratory epithelium. Similarly, neither TNFα nor the anti-inflammatory Interleukin 4 (IL-4) affected OCTN2 expression [115]. In another paper, LPS treatment has been found to induce the downregulation of the expression of both OCTN1 and OCTN2 mRNA and proteins with a concomitant increase in inflammation in the alveolar epithelial cell model of the A549 cells [116].

#### 4.3.2. Gut Epithelium

The dextran sodium sulfate (DSS)-induced colitis model is widely used for the study of IBD pathogenesis because this model represents many similarities to the immunological and histopathological features of human IBD. PPARα, PPARγ, and RXRα mRNA levels were downregulated in the colon of DSS-treated mice. In DSS-induced colitis, the agonist of PPARγ, luteolin, stimulating PPARγ reduced the expression levels of IL-1β and IL-6 and increased both the mouse Octn2 mRNA and protein expression. Mouse Octn1 upregulation has been described in the apical membrane of small intestinal epithelial cells after DSS treatment [68] and refs. herein. The severity of intestinal inflammation in DSS-induced colitis was greater in OCTN1−/− mice than in wild-type mice.

PPARγ expression in colon epithelial cells is associated with intestinal microorganisms likely from the involvement of the TLR4. The enteric microbiome is responsible for the strong regulation of PPARγ activation and OCTN2 expression [104]. A computational approach has been employed to analyze the effects of gut microbiome composition on gene expression in intestinal epithelial cells. Then, to screen differentially expressed genes, microbiota-depleted mouse samples were compared to control mouse samples, and OCTN1 resulted as enriched in the samples of IBD [117].

The effect of the proinflammatory cytokines TNF-α, IFN-γ, and IL-1β on OCTN2 expression was investigated in the human colon cell line FHC cells [67]. Mixed proinflammatory cytokines TNF-α, IL-1β, and IFNγ downregulated the expression of OCTN2 and reduced the carnitine content acting by PPARγ/RXRα pathways in FHC cells. Oppositely, in a further study, in the case of intestinal inflammation, IFN-γ was responsible for the stimulation of OCTN2 mRNA and protein expression along with increased total protein expression in the proximal colon and also increased apical OCTN2 abundance. TNFα does not alter colonic OCTN2 expression or activity but increases apical abundance and OCTN2 activity in the small intestine [118]. IFN-γ and TNFα increased the carnitine uptake. A decrease in the mRNA expression of OCTN2 was reported in the sigmoid region of the UC colon [119]. In a further study, the mRNA expression levels of OCTNs were detected in the terminal ileum and the colon of the intestine in IBD and UC patients [120]: in the terminal ileum of IBD patients, the mRNA expression levels of OCTN2 were significantly decreased, whereas changes in OCTN1 were not statistically significant. In the colon of IBD patients, OCTN2 mRNA levels were also significantly decreased. In the case of the mRNA expression of OCTN1, a significant increase was only observed in UC patients. The whole mucosal mRNA expression of OCTN1 was not significantly different between inflamed and non-inflamed mucosa, both in CD and UC patients. In contrast, the whole expression level of OCTN2 mRNA at the inflamed mucosa was significantly reduced compared to non-inflamed areas, both in CD and UC patients [121].

#### 4.3.3. Other Epithelia

In the placenta, the significant endogenous expression of TNFα has been detected during the first trimester of pregnancy; indeed, inflammation accompanies pregnancy for physiological reasons. OCTN1 and OCTN2 expression in placenta have been found [6]. This information would allow us to hypothesize a possible modulation in the mRNA expression of the OCTNs.

The effects of inflammation in the lactating mammary gland have been investigated, as well, by evaluating LPS-induced inflammation effects in the lactating rat mammary gland at different lactation stages [122]. The mRNA expression levels of OCTN1 were markedly higher in mammary glands at lactation day 11 compared to lactation day 4, though no statistical significance was observed. LPS downregulated OCTN2.

## 5. Relationships of OCTNs with Altered Metabolite Profiles in Inflammation-Based Diseases

In the inflammation-based pathologies, an alteration of OCTNs would correspond to altered protein activity or expression levels, resulting in an alteration of substrate concentration in tissues and body fluids. On this basis, studies that investigated the metabolite profiles of subjects affected by inflammation-based pathologies have been analyzed to identify the possible derangement of OCTN substrates.

A complex endogenous inflammatory cascade accompanies injury and repair processes after stroke. A metabolomic analysis of patient plasma samples collected multiple times after stroke reveals that five carnitine derivatives, and hence potential OCTN substrates, showed a gradually decreased concentration. The finding correlated well with the increase in ischemia’s energy requirement, which may cause a time-dependent depletion of acylcarnitines [123]. Moreover, other studies that compare carnitine and acylcarnitine abundance between patients with stroke and healthy subjects highlighted higher levels of these two metabolites in patient’s samples [124,125].

An alteration of potential OCTN substrates was also found in patients affected by RA, a chronic, inflammatory autoimmune disease characterized by joint inflammation, pain, and swelling, leading to cartilage and bone damage. For example, a recent study finds that erythrocyte ergothioneine concentrations are higher in patients with mild RA disease activity than in healthy individuals [126]. Moreover, in the case of RA, the concentrations of the circulating metabolites are modulated depending, on the one hand, on their release from the inflamed joint, and on the other hand, on their uptake by the synovium [127], thus highlighting the potential involvement of the transporters’ activities in determining the RA metabolomic profile. In particular, macrophages, T cells, and the fibroblast-like synoviocytes (FLSs), key cells involved in the pathogenesis and progression of RA, potentially release metabolites into the bloodstream. However, the evaluation of metabolic profile modifications in RA is not easy; indeed, it is influenced by many factors like food, drugs, the microbiome, etc. An increase in the abundance of bacteria responsible for TMAO production was found in new-onset untreated RA patients, and the metabolites related to the choline pathway were found in several studies in synovial tissue, synovial fluid, and blood (serum/plasma) samples in both animal models and humans. TMAO, as well as choline, was found to be increased in serum samples in the murine K/BxN model of arthritis compared to control mice as well [127]. Conversely, other studies conducted using rats with Collagen-Induced Arthritis (CIA), an autoimmune disease model that shares features with RA, showed reduced levels of choline in CIA rats compared to the control [128]. A decrease in plasma or blood choline levels was also described in studies that recruited human RA patients [129,130]. Choline decrease may be due to an increased choline uptake and consumption by the inflamed synovium [130,131]. Another study focused on the identification of the differentially abundant metabolites between higher and lower RA activity patients showed that among the 31 metabolites that increased in lower disease activity, seven (3-hydroxydecanoylcarnitine, dihomo-linoleoylcarnitine, eicosenoylcarnitine, linoleoylcarnitine, linoleoylcarnitine, stearoylcarnitine, and palmitoylcarnitine) are a part of acylcarnitine metabolism [132]. Moreover, the role of carnitine in the increased level of CCL20 in RA was suggested. Indeed, fatty acid oxidation, as well as glycolysis, has been implicated in the immune regulation and activation of macrophages. It has been hypothesized that the exposure of monocytes to the hypoxic and inflammatory RA environment can impact their metabolic state and it would suggest that the increased carnitine abundance is part of a hypermetabolic state that can drive a CCL20-mediated inflammatory cascade to promote disease pathogenesis [133,134].

Osteoarthritis is another joint disease with chronic inflammation, progressive articular cartilage destruction, and subchondral bone sclerosis. Inflammation drives chondrocytes to express ECM-degrading enzymes, and the interruption of this pathway is a viable target to prevent cartilage degradation. It has been demonstrated that inflammation can alter the intracellular metabolism of chondrocytes, a process known as metabolic reprogramming [135]. Carlson et al. detected 1233 metabolites in the synovial fluid (SF), representative of the most accessible tissue near chondrocytes. In these samples, 35 potential biomarkers of OA, including phosphatidylcholine, lysophosphatidylcholine, and carnitine derivatives, have been identified [136]. Interestingly, a previous study conducted using a targeted metabolomics approach to identify metabolic markers for different class osteoarthritis patients, found that acylcarnitine and free carnitine may be potentially useful in distinguishing the different OA subtypes by analyzing SF samples [137]. Moreover, Mickiewicz et al. found reduced levels of acetylcarnitine, hexanoylcarnitine, N-phenylacetylglycine, and ethanolamine in OA samples compared to the controls [138]. Similarly, Tootsi et al. describe decreased medium- and long-chain acylcarnitines associated with OA severity [139]. The same study suggested that acylcarnitines might be important in the link between OA and cardiovascular comorbidity.

Another set of pathologies characterized by inflammation is IBD. The alteration of OCTN substrates/potential substrates in IBD patients has been found [140]. On the contrary, acylcarnitines appear to be more abundant in the CD patients compared with healthy controls. Interestingly, the abundance of certain types of acylcarnitines can help to classify IBD, IBD with damp-heat syndrome (IBD-DH), and IBD with spleen deficiency syndrome (IBD-SD). Indeed, Wu et al. demonstrated that three specific acylcarnitines (ACar 20:4, ACar 18:1, and ACar 20:3) were significantly increased in IBD plasma samples and that the ACar 8:1 was significantly increased in IBD with damp-heat syndrome when compared with IBD with spleen deficiency syndrome [141]. Another study found that the fecal acylcarnitine content in patients with IBD and mice with experimental colitis tended to increase compared to the control group, and the reason may be related to intestinal inflammation that led to mitochondrial dysfunction in the apical domain of the surface epithelium that may reduce the consumption of fatty acids [86]. Similar results were obtained from a fecal metabolome study on 424 patients with IBD and 255 non-IBD controls [142]. Moreover, the prediction of the disease course is highly desirable. This goal could be achievable using a multi-omics approach on serum samples; indeed, an established model of four metabolites, propionyl-L-carnitine, carnitine sarcosine, and sorbitol, combined with three proteins, IL-10, glial cell line-derived neurotrophic factor, and the T-cell surface CD8 alpha chain, was found to be predictive of relapse within two years [143].

Chronic inflammation has also been shown to play a role in the pathogenesis of frailty, a geriatric syndrome. The age-related significant rise in inflammatory markers, “inflammaging”, could predispose to frailty. The dysregulation of the carnitine shuttle has been found to play a role in the risk of frailty. Indeed, a frailty metabolic phenotype, including a decrease in six carnitines, distinguishes frail from non-frail phenotypes [144]. In the same work, an SNP for OCTN1 has been recently associated with decreased carnitine levels in frailty.

## 6. Perspectives and Concluding Remarks

The link between OCTNs and inflammatory diseases emerged in the last years. The ubiquitous tissue expression of these transporters together with the increasing numbers of studies on their regulation indicate good perspectives of deeply understanding their role in inflammation. Hence, due to the complexity of metabolic phenomena occurring in inflammation, future studies on OCTNs will furnish an important piece of knowledge in dissecting the molecular aspects of metabolic alterations occurring in inflammatory processes. Moreover, the increasing refinement of the structural aspects of these transporters and their structure/function relationships will give great support for the design of novel drugs as well as for the appropriate repurposing of already-known drugs for controlling deleterious aspects of inflammation.

## Figures and Tables

**Figure 1 biomolecules-14-00392-f001:**
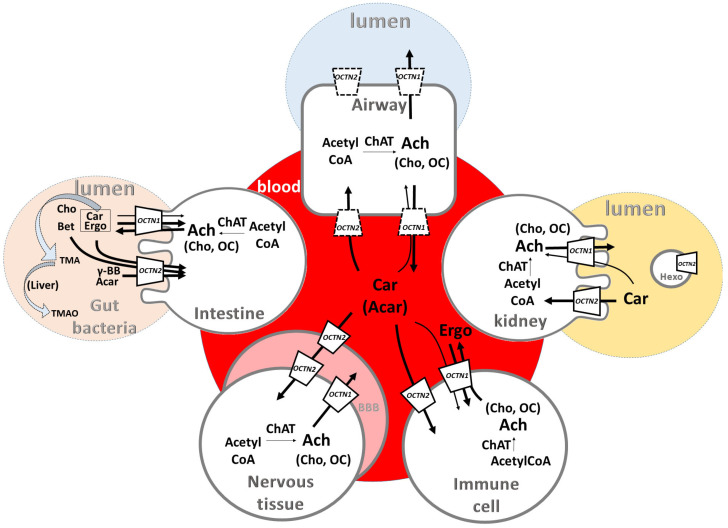
Major tissue localization of OCTNs and role in transport of inflammation-related metabolites. Intestine epithelial cell on left and kidney epithelial cell on right are depicted with brush border membranes; airway cell on top, nervous tissue and immune cells at bottom; blood–brain barrier (BBB) is represented as pink barrier. Exo, exosome. Sketch of some human or gut bacteria pathways are depicted: TMA, trimethylamine; TMAO, trimethylamine oxide; ChAT, choline acetyltransferase. OCTNs in dotted lines represent controversial localization [20,21]; Ergo, ergothioneine; Ach, acetylcholine; Cho, choline; Acar, acetylcarnitine; Bet, betaine; γ-BB, γ-butyrobetaine; OC, organic cation. Arrows indicate transport direction. Thin arrows refer to low-affinity transport.

**Figure 2 biomolecules-14-00392-f002:**
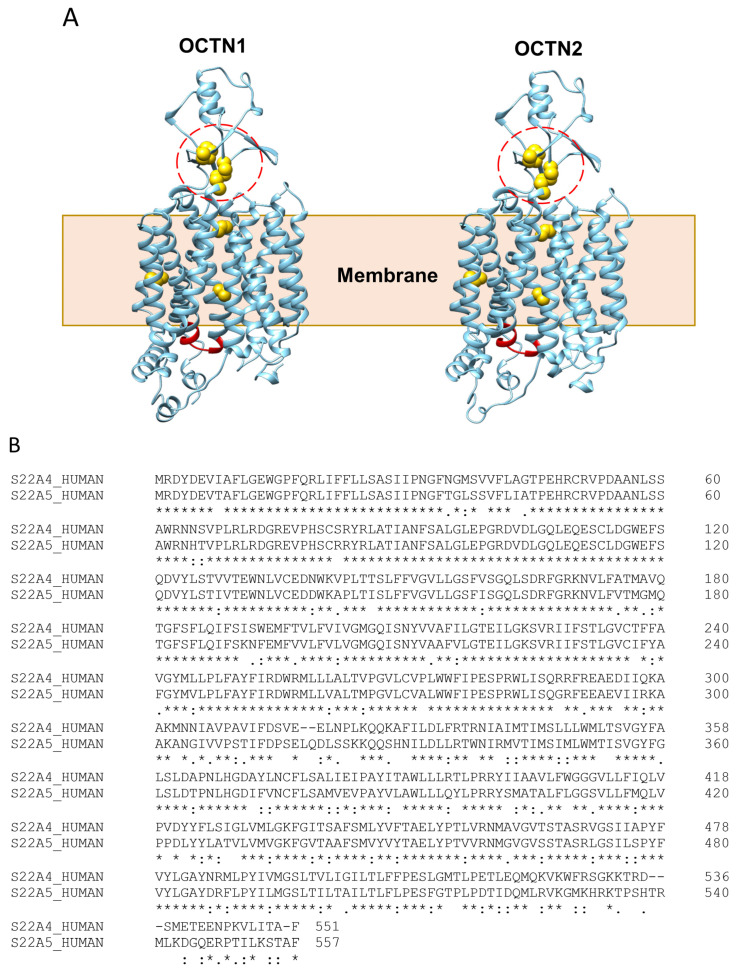
A comparison of OCTNs’ structural features. (**A**) The lateral view of the AlphaFold structures of human OCTN1 (SLC22A4) and OCTN2 (SLC22A5) are depicted in light blue using a ribbon representation. Cysteine residues are highlighted in yellow using a space-filled representation; the dotted red circles highlight the four cysteines in the extracellular loop. The putative ATP binding site is shown in red. (**B**) Amino acid sequences of human OCTNs. Conserved amino acid residues among members are indicated by stars.

**Figure 3 biomolecules-14-00392-f003:**
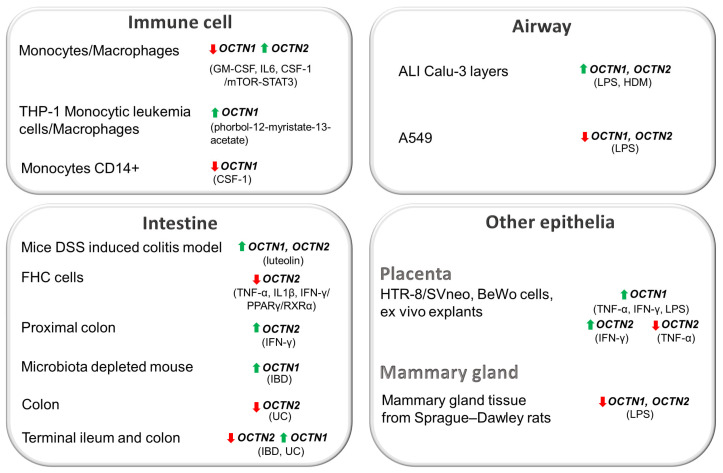
Modulation of OCTN expression. Cells of indicated tissues are surrounded by grey boxes. Arrows indicate gene/protein upregulation (green) or downregulation (red).

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
