# Peer review of "Inflammation and Organic Cation Transporters Novel (OCTNs)"

_biomolecules, 2024, doi:10.3390/biom14040392_

Round 1

Reviewer 1 Report

Comments and Suggestions for Authors

Pochini et al. provide a comprehensive review about the role of the OCTN transporters OCTN1 and OCTN2 in different inflammation-associated diseases. The review is well structure and nicely written. I have only some minor points that should be considered for revision.

1. For readers that are not well involved in the OCTN field, Figure 1 is difficult to understand. Different transport modes are presented that are not further describe in the rieview. I recommend to include a paragraph about the transport mode of OCTN1 and OCTN2.

2. Figure 2: Homology models are mentioned on page 3, lines 85ff, but the figure shows AlphaFold structures. This should be explained.

3. The review would benefit from an additional figure summarizing the regulation of OCTN gene expression.

- Figure 2: Please use the protein names OCTN1 and OCTN2 instead of the gene names (SLC22A4, A5) as the proteins are shown.

- Page 4, line 141: not only humans!

- Include Supplementary Table 1 as a regular Table.

- Speeling. Please be more consistent with using abbreviations and be more case sensitive. Please introduce an abbreviation at the first use and then consequently use the abbreviation (e.g. for inflammatory bowel  disease, IBD; Ulcerative Colitis, UC; Rheumatoid Arthritis, Rheumatoid arthritis, RA). Please use lower case if no proper noun is meant (e.g. Choline, Peroxisome proliferation-activyted receptor, Peroxisome Proliferation Responsive Elements).

- Please explain that for humans you use upper case (OCTN) and for animals lower case (octn).

- Page 3, line 89: it shold be stated that the intracellular nucleotide-binding domain is different to the NBDs of ABC transporters.

- Page 2, line 52: identity to each other

- Page 2: line 65: intestine epithelial cells and kidney epithelial cells

- Page 9, line 407: ulcerative colitis

Author Response

Pochini et al. provide a comprehensive review about the role of the OCTN transporters OCTN1 and OCTN2 in different inflammation-associated diseases. The review is well structure and nicely written. I have only some minor points that should be considered for revision.

  1. For readers that are not well involved in the OCTN field, Figure 1 is difficult to understand. Different transport modes are presented that are not further describe in the rieview. I recommend to include a paragraph about the transport mode of OCTN1 and OCTN2.

R.1. We agree with the reviewer comment; thus, we have inserted a paragraph on the transport mode (lines 76-79).

  1. Figure 2: Homology models are mentioned on page 3, lines 85ff, but the figure shows AlphaFold structures. This should be explained.

R.2. We thank the reviewer for this comment. We have corrected the text that was misleading (lines 89-92)

  1. The review would benefit from an additional figure summarizing the regulation of OCTN gene expression.

R.3. We have added a new figure 3 that, indeed, is useful for facilitating the readers.

- Figure 2: Please use the protein names OCTN1 and OCTN2 instead of the gene names (SLC22A4, A5) as the proteins are shown.

  1. Done

- Page 4, line 141: not only humans!

  1. The sentence has been modified by deleting “for humans” (line 151)

- Include Supplementary Table 1 as a regular Table.

  1. In principle we agree with the reviewer; however, since the table is a systematic list of polymorphisms, it is more than 7 pages long and will cause a too long interruption to the text of the manuscript that may impair readability, therefore, we have preferred to leave the table as supplementary.

- Speeling. Please be more consistent with using abbreviations and be more case sensitive. Please introduce an abbreviation at the first use and then consequently use the abbreviation (e.g. for inflammatory bowel disease, IBD; Ulcerative Colitis, UC; Rheumatoid Arthritis, Rheumatoid arthritis, RA). Please use lower case if no proper noun is meant (e.g. Choline, Peroxisome proliferation-activyted receptor, Peroxisome Proliferation Responsive Elements).

  1. Changes suggested by the reviewer have been performed throughout the whole text.

- Please explain that for humans you use upper case (OCTN) and for animals lower case (octn).

  1. The text has been modified to make clear the use of “octn2 for animals, in particular for mouse (Line 406).

- Page 3, line 89: it shold be stated that the intracellular nucleotide-binding domain is different to the NBDs of ABC transporters.

  1. Text has been modified according to the reviewer suggestion (lines 94-95).

- Page 2, line 52: identity to each other

  1. Done

- Page 2: line 65: intestine epithelial cells and kidney epithelial cells

  1. Done

- Page 9, line 407: ulcerative colitis

  1. Corrected (line 423)

Reviewer 2 Report

Comments and Suggestions for Authors

In this review, the authors discuss two membrane transporters - OCTN1 (SLC22A4) and OCTN2 (SLC22A5) – focusing on their transport function and regulation in inflammation or by inflammatory diseases. The authors have solid expertise in this field and published several original papers and reviews on these proteins. The review is informative, clearly written and interesting to read.

The following issues can be improved before the manuscript is published.

1.  Authors can add some general sentences about the SLC22 family and other OCTNx at the beginning of the review.

2. Larger font in Figure 1 will make the figure easier to read.

3. The addition of the homology of the primary structures of OCTN1 and OCTN2 in Figure 2 would be useful.

4. Line 108. How specific are the antibodies used in the studies of OCTN expression?

5. Line 338-342 Since it is possible to polarize monocytes into anti- or proinflammatory macrophages, it would be interesting to know, whether the expression of OCTN1 and OCTN2 differs in different macrophage populations.

Comments on the Quality of English Language

English should be improved throughout the manuscript.

Some examples:

Line 52: identity each other à identity to each other,

Line 92: Cysteines à cysteines

Lines 127: relationships among between inflammatory processes and the OCTNs

Line 130: Relationships among  between functions and diseases

Line 147: among which OCTN1 à among which is OCTN1; It has been known for 20 years

Line 181: This is associated to with Crohn's disease

Line 291: Inflammation à inflammation

Author Response

In this review, the authors discuss two membrane transporters - OCTN1 (SLC22A4) and OCTN2 (SLC22A5) – focusing on their transport function and regulation in inflammation or by inflammatory diseases. The authors have solid expertise in this field and published several original papers and reviews on these proteins. The review is informative, clearly written and interesting to read.

The following issues can be improved before the manuscript is published.

  1. Authors can add some general sentences about the SLC22 family and other OCTNx at the beginning of the review.

R.1. Some general information on SLC22 and, in particular, cation transporters have been added (lines 47-51).

  1. Larger font in Figure 1 will make the figure easier to read.

R.2. Done

  1. The addition of the homology of the primary structures of OCTN1 and OCTN2 in Figure 2 would be useful.

R.3. Done, (please see new fig 2B) and legend has been modified (lines 101-106)

  1. Line 108. How specific are the antibodies used in the studies of OCTN expression?

R.4. We have cited many studies on OCTN expression, concurring to similar results. Anyway, in our experience, the antibody produced by several companies against OCTNs are quite specific.

  1. Line 338-342 Since it is possible to polarize monocytes into anti- or proinflammatory macrophages, it would be interesting to know, whether the expression of OCTN1 and OCTN2 differs in different macrophage populations.

R.5. We thank the reviewer for the interesting suggestion. We will consider this issue in our further studies on OCTNs.

English should be improved throughout the manuscript.

Some examples:

Line 52: identity each other à identity to each other,

Line 92: Cysteines à cysteines

Lines 127: relationships among between inflammatory processes and the OCTNs

Line 130: Relationships among between functions and diseases

Line 147: among which OCTN1 à among which is OCTN1; It has been known for 20 years

Line 181: This is associated to with Crohn's disease

Line 291: Inflammation à inflammation

  1. All suggested corrections have been made and the manuscript has been further subjected to language editing.

Reviewer 3 Report

Comments and Suggestions for Authors

Overall

Quite a good review. There are some issues with depth and explanation of the information provided on OCTNs (see specific points below) and also some suggestions on the structure of the review. One major point is that there are very few Figs./tables for such a long review. The authors might strongly consider providing diagrams for some of the major points made, including:

i) the hypothesis that OCTNs are involved in gut inflammation via excess TMAO production in OCTN functional mutants or down-/up--regulated in immunological models of gut inflammation (sections 3.2 and 4.3.2). These sections together provide a lot of information without and synthesis and summary/overview by the author (see note 19 below). 

ii) a schematic summarising the regulatable elements in OCTN1 and 2 loci (section 4.1), indicating whether the binding region has been demonstrated with experimental results or if it simple annotated bioinformatically. 

iii) OCT1 and 2 role in RA via transcriptional regulation via RUNX1 (section 4.1)

iv) the changes in OCTN1 and 2 expression during monocyte to macrophage activation (section 4.2)

Specific Points:

1. Title: should use the full name of the OCTNs, Novel Organic Cation Transporters

2. Line 50: SLC needs to be written out fully at the first usage and what the classification system is outlined (i.e. Human Genome codes).

3. Line 77: 'diet-microbiome axis' many readers will have some understanding of this term but it needs to be more fully explained (in general terms) how the axis can condition circulating metabolites.

4. Line 79: 'unknown substrates' how can you conclude there are unknown substrates if they are, indeed, unknown? Is there direct evidence for specific metabolites and a rationale that allows for the possibility that such metabolites may be OCTN substrates? If used you need to expand on this assertion with details somewhere. The same point holds for the use of 'potential' substrates of OCTNs on line 82: what is the rationalisation for designating particular metabolites as 'potential' substrates - these need fleshing out.

5. Line 84: by 'described similarity' do you mean similar substrate specificity? It is unclear what similarity you are referring to. 

6. Line 88: N- or O-linked glycosylation sites?

7. In fig. 2 you are using the alpha-fold structures, yet in the text you refer to homology models of OCTN1 and 2. The biochemical and structural information you provide in the text is derived from which ones? Please be specific and consistent. 

8. Line 106: could you expand on the information provided here, it is rather vague: is there any evidence of a direct link between ROS and OCTNs that link to the ICTN rile in inflammation? In other words, and as a more general point, please provide a summary of the literature evidence that is specific and clear. Is there evidence or is this simply your hypothesis. 

9. Title to section 2.1: you mean 'between' rather than 'among', I think?

10. A general point on the structure of the review from section 2: it seems a little disjointed to group the basic biochemistry and function of the OCTNs with disease conditions in section 2.1. Would the revie not be better structured to first give the overview of OCTN biochemistry (specificity, kinetics, tissue expression, localisation, transport mechanism) before you give an in-depth review of OCTN role in inflammation? Some this basic information is missing (e.g. stoichiometry, tissue expression) and it would also help you explain how OCTN plays a role in the aetiology. In short, section 2.1 seems incomplete.

11. Line 183: What are IBDs? Please spell out acronyms when first used. The same with 'iNOS' on line 216 and TMAO (line 259). Same again wit PCD (line 270) and ChAT (line 282). Please do this for all acronyms not named at first use or provided a list.  

12. Line 186 to 196: where are the electronic reference for these database SNP figures and the disease-causing mutation figures? How is known that 145 protein variants are 'clearly pathogenic'? No evidence is provided in the form of references. If these references are all contained in Table S1 (as I suspect they are) please make this clear to the reader in the text.

13. Line 206: this is interesting, but please detail the specific evidence from these references that demonstrates the biochemistry behind OCTN, its substrate ergothioneine and IL-1β. A good example of the level of detail that should be provided for all information you introduce is in the very next paragraph, showing the role of carnitine deficiency, OCTN2 and inflammation responses.

14. Line 254: 'frontier' of what?

15. Line 265: where specifically was the concentrative cooperation of OCTN1 and 2 described? Help the reader locate it more easily or describe it briefly here.

16. Section 3.2 is very interesting but the main point was lost somewhat in the text. If I understand correctly, only mutations in OCTNs which inhibited their normal intestinal absorption capacity for carnitine would likely to increase bacterial TMAO in the gut? you should expand on this hypothesis and make it clear to the reader.

17. The last paragraph of section 3.2 is unlcear and the link to OCTNs, if any, I cannot see. Is the evidence that propionate (produced by the microbiota) binding to GPR41/GPR43 stimulated acetylcholine release? If so, where is the link to inflammation? And what is the role of OCTNs here? Is the acetylcholine release facilitated by OCTNs? At the moment it simply seems like acetylcholine (which happens to be a OCTN substrate) is essential in this process but there is not role for OCTNs. If that is the case why is this even in the review? This section needs to be clarified and written more clearly.

18. Line 395: this sentence is grammatically incorrect. I cannot understand it.

19. Why is section 4.3.2 separated from section 3.2: both deal with inflammation in the gut epithelium? This especially true as in section 3.2 you posit a proinflammatory role for OCTNs via carnitine uptake, yet in 4.3.2 evidence suggest, at least octn1, plays a role in ameliorating the severity of IBD inflammation. And yet again, OCTN2 expression is down regulated or upregulated (depending on the studied? It is unclear how this various regulation would work in concert. I think this evidence needs more coherence in its explanation. Would you like to expand and explain these discrepancies, if I have understood the data correctly? And give a stronger author analysis and interpretation. This would be well done under a single section.

20. Line 383: where are the references providing evidence that the DSS-induced model of IBD retains the immunological features of the disease that allow it to be a model for study?

21. Line 423: you provide no references for this information.

Comments on the Quality of English Language

English largely fine. The odd grammatical mistake needs correcting by a careful reading. 

Author Response

Overall

Quite a good review. There are some issues with depth and explanation of the information provided on OCTNs (see specific points below) and also some suggestions on the structure of the review. One major point is that there are very few Figs./tables for such a long review. The authors might strongly consider providing diagrams for some of the major points made, including:

  1. i) the hypothesis that OCTNs are involved in gut inflammation via excess TMAO production in OCTN functional mutants or down-/up--regulated in immunological models of gut inflammation (sections 3.2 and 4.3.2). These sections together provide a lot of information without and synthesis and summary/overview by the author (see note 19 below). 

R.i). We acknowledge that section 3.2 lacks a synthesis of the important findings described. Therefore, we have added a final paragraph (lines 307-309).

  1. ii) a schematic summarising the regulatable elements in OCTN1 and 2 loci (section 4.1), indicating whether the binding region has been demonstrated with experimental results or if it simple annotated bioinformatically. 

iii) OCT1 and 2 role in RA via transcriptional regulation via RUNX1 (section 4.1)

  1. iv) the changes in OCTN1 and 2 expression during monocyte to macrophage activation (section 4.2)

R.ii-iv. We have inserted a new figure (fig 3 and related legend) summarizing some of the clearest regulation mode of OCTN1 and 2 (see also response R.3 to reviewer 1).

Specific Points:

  1. Title: should use the full name of the OCTNs, Novel Organic Cation Transporters

R.1. Done.

  1. Line 50: SLC needs to be written out fully at the first usage and what the classification system is outlined

R.2. Done (lines 48-49)

  1. Line 77: 'diet-microbiome axis' many readers will have some understanding of this term but it needs to be more fully explained (in general terms) how the axis can condition circulating metabolites.

R.3. Explanation has been provided (lines 81-83)

  1. Line 79: 'unknown substrates' how can you conclude there are unknown substrates if they are, indeed, unknown? Is there direct evidence for specific metabolites and a rationale that allows for the possibility that such metabolites may be OCTN substrates? If used you need to expand on this assertion with details somewhere. The same point holds for the use of 'potential' substrates of OCTNs on line 82: what is the rationalisation for designating particular metabolites as 'potential' substrates - these need fleshing out.

R.4. The reviewer is right, so we have changed the text adding possibly “unknown” in view of the polispecificity of these transporters: line 87

  1. Line 84: by 'described similarity' do you mean similar substrate specificity? It is unclear what similarity you are referring to. 

R.5. Text has been modified to improve clarity (line 90)

  1. Line 88: N- or O-linked glycosylation sites?

R.6. Text has been modified to complete the description (line 93)

  1. In fig. 2 you are using the alpha-fold structures, yet in the text you refer to homology models of OCTN1 and 2. The biochemical and structural information you provide in the text is derived from which ones? Please be specific and consistent. 

R.7. We have clarified this issue (lines 89-92).

  1. Line 106: could you expand on the information provided here, it is rather vague: is there any evidence of a direct link between ROS and OCTNs that link to the ICTN rile in inflammation? In other words, and as a more general point, please provide a summary of the literature evidence that is specific and clear. Is there evidence or is this simply your hypothesis. 

R.8. We have clarified some results (line 112-113). The link with physiological ROS remains a hypothesis.

  1. Title to section 2.1: you mean 'between' rather than 'among', I think?

R.9. Corrected

  1. A general point on the structure of the review from section 2: it seems a little disjointed to group the basic biochemistry and function of the OCTNs with disease conditions in section 2.1. Would the revie not be better structured to first give the overview of OCTN biochemistry (specificity, kinetics, tissue expression, localisation, transport mechanism) before you give an in-depth review of OCTN role in inflammation? Some this basic information is missing (e.g. stoichiometry, tissue expression) and it would also help you explain how OCTN plays a role in the aetiology. In short, section 2.1 seems incomplete.

R.10. We agree with the reviewer. However, to not extend too much this section, we have referred to other papers containing the information.

  1. Line 183: What are IBDs? Please spell out acronyms when first used. The same with 'iNOS' on line 216 and TMAO (line 259). Same again wit PCD (line 270) and ChAT (line 282). Please do this for all acronyms not named at first use or provided a list.  

R.11. Done (see also response to reviewer 1)

  1. Line 186 to 196: where are the electronic reference for these database SNP figures and the disease-causing mutation figures? How is known that 145 protein variants are 'clearly pathogenic'? No evidence is provided in the form of references. If these references are all contained in Table S1 (as I suspect they are) please make this clear to the reader in the text.

R.12. Thank you for this observation. We have inserted a sentence to make clear this point (lines 207-208).

  1. Line 206: this is interesting, but please detail the specific evidence from these references that demonstrates the biochemistry behind OCTN, its substrate ergothioneine and IL-1β. A good example of the level of detail that should be provided for all information you introduce is in the very next paragraph, showing the role of carnitine deficiency, OCTN2 and inflammation responses.

R.13. We have better explained the effect (lines 219-220).

  1. Line 254: 'frontier' of what?

R.14. Text has been modified to better explain this concept (line 268).

  1. Line 265: where specifically was the concentrative cooperation of OCTN1 and 2 described? Help the reader locate it more easily or describe it briefly here.

R.15. Text has been modified to better explain this concept (line 280).

  1. Section 3.2 is very interesting but the main point was lost somewhat in the text. If I understand correctly, only mutations in OCTNs which inhibited their normal intestinal absorption capacity for carnitine would likely to increase bacterial TMAO in the gut? you should expand on this hypothesis and make it clear to the reader.

R.16. We thank the reviewer for this observation. We have clarified the raised point (lines 285-286).

  1. The last paragraph of section 3.2 is unlcear and the link to OCTNs, if any, I cannot see. Is the evidence that propionate (produced by the microbiota) binding to GPR41/GPR43 stimulated acetylcholine release? If so, where is the link to inflammation? And what is the role of OCTNs here? Is the acetylcholine release facilitated by OCTNs? At the moment it simply seems like acetylcholine (which happens to be a OCTN substrate) is essential in this process but there is not role for OCTNs. If that is the case why is this even in the review? This section needs to be clarified and written more clearly.

R.17. We thank the reviewer for the observation; we have clarified this section (lines 298, 298-300).

  1. Line 395: this sentence is grammatically incorrect. I cannot understand it.

R.18 We thank the reviewer for the observation; we corrected the sentences (lines 412-416)

  1. Why is section 4.3.2 separated from section 3.2: both deal with inflammation in the gut epithelium? This especially true as in section 3.2 you posit a proinflammatory role for OCTNs via carnitine uptake, yet in 4.3.2 evidence suggest, at least octn1, plays a role in ameliorating the severity of IBD inflammation. And yet again, OCTN2 expression is down regulated or upregulated (depending on the studied? It is unclear how this various regulation would work in concert. I think this evidence needs more coherence in its explanation. Would you like to expand and explain these discrepancies, if I have understood the data correctly? And give a stronger author analysis and interpretation. This would be well done under a single section.

R.19. Section 4.3.2 deals with regulation of OCTNs in gut epithelium and, thus, it has been separated by the previous section (3.2) dealing with substrates. However, we agree that the reader may have some confusion, therefore, also for responding to reviewer 1, we have added a figure summarizing the regulation of the two transporters in different tissues. Moreover, the reviewer is right that the description of expression in IBD is not clear. We have rewritten this part (lines 428-432).

  1. Line 383: where are the references providing evidence that the DSS-induced model of IBD retains the immunological features of the disease that allow it to be a model for study?

R.20. We thank reviewer and we added “refs herein” (line 408), since ref 68 includes several other references dealing with the same issue.

  1. Line 423: you provide no references for this information.

R.21. Text has been modified and a reference added (lines 440-442)